# Evocalcet with vitamin D receptor activator treatment for secondary hyperparathyroidism

**Takashi Shigematsu**[1]*, **Shinji Asada**[2], **Yuichi Endo**[3], **Takehisa Kawata**[2], **Masafumi Fukagawa**[4], **Tadao Akizawa**[5]

**1** Department of Nephrology, Wakayama Medical University, Wakayama, Japan, **2** Medical Affairs Department, Kyowa Kirin Co., Ltd., Tokyo, Japan, **3** R&D Division, Kyowa Kirin Co., Ltd., Tokyo, Japan, **4** Division of Nephrology, Endocrinology, and Metabolism, Department of Internal Medicine, Tokai University School of Medicine, Kanagawa, Japan, **5** Division of Nephrology, Department of Medicine, Showa University School of Medicine, Tokyo, Japan

* docteur_shigemat@iris.eonet.ne.jp

**Data Availability Statement:** All relevant data are within the manuscript and its Supporting Information files.

## Abstract

This *ad hoc* analysis of a previously conducted phase 3 head-to-head comparison study of evocalcet and cinacalcet in secondary hyperparathyroidism patients undergoing maintenance hemodialysis evaluated the efficacy and safety of combined once-daily oral evocalcet and intravenous vitamin D receptor activator treatment stratified by weekly vitamin D receptor activator dose (117, 45, and 91 patients in no, low [< 1.5 μg], and high [≥ 1.5 μg] dose groups, respectively). Effects of vitamin D receptor activator were assessed on the basis of intact parathyroid hormone, corrected calcium, phosphorus, and fibroblast growth factor-23 levels; percent changes from baseline; proportions of patients who achieved target intact parathyroid hormone, corrected calcium, and phosphorus at Weeks 28–30; and adverse drug reactions. Intact parathyroid hormone, corrected calcium, phosphorus, and fibroblast growth factor-23 levels decreased in all groups; phosphorus and fibroblast growth factor-23 levels remained high in the high dose group. In the low and high dose groups, greater proportions of patients achieved the corrected calcium target compared with the no dose group ($p = 0.043$). Ratios of intact-to-C-terminal fibroblast growth factor-23 decreased in all groups. In low and high dose groups, hypocalcemia was less common than in the no dose group ($p = 0.014$). Evocalcet with concomitant vitamin D receptor activator demonstrated benefits such that more patients achieved the corrected calcium target and exhibited decreased fibroblast growth factor-23 synthesis; the incidence of hypocalcemia also decreased.

**Clinical trial registration**: ClinicalTrials.gov (NCT02549391) and JAPIC (JapicCTI-153013).

## Introduction

Patients with severe chronic kidney disease requiring dialysis exhibit a decline in renal function that controls calcium and phosphate metabolism. This impairment subsequently causes

**Funding:** The study was funded by Kyowa Kirin Co., Ltd. The funder provided support in the form of consultancy fees and lecture fees to authors TS, MF and TA, salaries to authors SA, YE, and TK and grant to author MF. The funder was involved in the study design, data collection and analysis, decision to publish, and preparation of manuscript.

**Competing interests:** TS received consulting fees from Kyowa Kirin Co., Ltd. (KKC), Ono Pharmaceutical, Taisho Toyama Pharmaceutical, Fuji Pharma, and FUSO, and lecture fees from KKC, Chugai Pharmaceutical, Bayer, Kissei Pharmaceutical, Torii Pharmaceutical, Ono Pharmaceutical, and FUSO. SA, YE, and TK are employees of KKC. MF received consulting fees from KKC and Ono Pharmaceutical; lecture fees from KKC, Bayer, Torii Pharmaceutical, and Ono Pharmaceutical; and grants from KKC and Bayer. TA received consulting fees from KKC, Astellas Pharma, Bayer, Fuso Pharmaceutical, Japan Tobacco, Ono Pharmaceutical, Sanwa Chemical, Otsuka, GSK, and NIPRO, and lecture fees from KKC, Chugai Pharmaceutical, Bayer, Kissei Pharmaceutical, Torii Pharmaceutical, and Ono Pharmaceutical. This does not alter our adherence to PLOS ONE policies on sharing data and materials.

various mineral and bone disorders, such as secondary hyperparathyroidism (SHPT) [1–3]. Patients with SHPT have elevated circulating parathyroid hormone (PTH), calcium and phosphorus levels, and increased levels of bone turnover markers and fibroblast growth factor-23 (FGF23), leading to an increased risk of cardiovascular morbidity and mortality [4].

FGF23, first reported in 2000, is synthesized in osteocytes and involved in the regulation of phosphorus reabsorption and $1,25(OH)_2$ vitamin D production [5, 6]. Recent studies revealed a significant association between elevated serum FGF23 levels and poor prognosis, cardiovascular events, and increased all-cause mortality in hemodialysis patients [7–12]. FGF23 is also associated with left ventricular hypertrophy in chronic kidney disease patients and, therefore, it is critical to control circulating FGF23, PTH, and mineral levels [13–15].

A vitamin D receptor activator (VDRA) preparation administered intravenously (IV) is the standard therapy for patients with SHPT in Japan, but recently, calcium-sensing receptor agonists, termed calcimimetics, have been developed and demonstrated a significant improvement in the balance between serum calcium, PTH, phosphorus, and FGF23 levels [16–25]. Evocalcet, the latest calcimimetic drug approved in Japan, causes nausea and vomiting less frequently than another calcimimetic drug, cinacalcet [26, 27], which suggests that it might better achieve target PTH and calcium levels in patients with SHPT who are on maintenance hemodialysis.

Because of the recent development of calcimimetics, patients with SHPT are often treated with combination therapy consisting of VDRA and a calcimimetic drug. Indeed, approximately 50% of patients who participated in a previously conducted phase 3 randomized study to compare the efficacy and safety of evocalcet with cinacalcet (phase 3 head-to-head comparison study) were treated with concomitant IV VDRAs [26]. Although VDRAs and calcimimetics may reduce PTH levels to a similar degree, they have different effects on calcium, phosphorus, and FGF23 levels [28–34]. Therefore, it is of clinical interest to investigate how the efficacies of various calcimimetics are affected by concomitant VDRA therapy.

The current study evaluated the efficacy, including that on FGF23, and safety of evocalcet in the presence (low and high VDRA use) or absence of concomitant IV VDRA using a data set obtained from the previous phase 3 head-to-head comparison study in patients with SHPT who were on hemodialysis.

## Materials and methods

This study was conducted at 89 study sites in Japan from October 2015 to November 2016. The ethics committees at all sites (S1 Table) approved the study protocol, and the study was conducted in accordance with the principles of the Declaration of Helsinki and in compliance with the Pharmaceuticals, Medical Devices, and Other Therapeutic Products Act, and Good Clinical Practice (Ministry of Health, Labour and Welfare Ordinance No. 28 dated March 27, 1997). The full study procedures were explained, and written informed consent was obtained from all patients prior to participation.

### Previous phase 3 head-to-head comparison study

**Patients.** Patients with SHPT and on maintenance hemodialysis were eligible for participation in the previous phase 3 head-to-head comparison study of evocalcet and cinacalcet [26]. The inclusion criteria were age ≥ 20 years, stable chronic renal failure treated with hemodialysis three times weekly for ≥ 12 weeks, mean intact PTH level > 240 pg/mL during screening at 2 weeks and 1 week before starting treatment, and serum corrected calcium (cCa) level ≥ 9.0 mg/dL during screening.

**Study procedure.** Detailed study procedures of the phase 3 head-to-head comparison study were described previously [26]. Briefly, the randomized, double-blind, intrapatient dose-adjustment, parallel-group study comprised a 28-week dose adjustment period (Week 0 to 1 day before Week 28) and a 2-week evaluation period (Week 28 to Week 30). During the dose adjustment period, the starting dose of once-daily oral evocalcet was determined based on the serum intact PTH level 1 week prior to treatment initiation: patients with < 500 pg/mL received 1 mg/day and those with ≥ 500 pg/mL received 2 mg/day that was further adjusted at 1 mg increments up to 8 mg on the day of dialysis based on the dose adjustment criteria.

Concomitant medication and therapy, including cinacalcet (from 2 weeks prior to screening to the end of the study), bisphosphonates, denosumab, teriparatide, and parathyroid intervention (from 24 weeks prior to screening to the end of study), and peritoneal dialysis (between Week 12 and Week 30), were prohibited. Changes in prescribed dialysis conditions (dialysate calcium concentration, dialyzer, dialysis time, and number of dialysis sessions per week) were prohibited from 2 weeks prior to screening to the end of the study.

During the period from 2 weeks before screening until the end of the study, initiation of or changes in preparation, dose, and dosing regimen of IV and oral VDRA medications or derivatives (calcitriol, maxacalcitol, falecalcitriol, alfacalcidol, and eldecalcitol) were not permitted. However, dose reduction or discontinuation of VDRA preparations and derivatives were allowed if the serum cCa level was > 11.0 mg/dL after the start of the study treatment. Furthermore, initiation of treatment with VDRA preparations and derivatives or any dose increase was permitted if either of the following conditions were met: corrected serum calcium level decreased below 7.5 mg/dL and remained at ≤ 7.5 mg/dL after the dose increase or initiation of a calcium preparation; or appearance of clinical symptoms potentially attributable to hypocalcemia that did not improve after the initiation or dose increase of a calcium preparation.

The initiation of phosphate binders, calcium preparations, or food with a phosphate binding effect was not allowed for patients who were not already receiving these preparations or food during this period.

## Study design and patients

This *ad hoc* analysis was designed to analyze the efficacy and safety results of the previous phase 3 head-to-head comparison study, stratified by the baseline weekly IV VDRA dose: patients in the evocalcet arm treated with no VDRA as no dose group (NDG); < 1.5 µg/week VDRA as low (L)DG; ≥ 1.5 µg/week VDRA as high (H)DG. The prescribed weekly dose of VDRA derivatives was converted to an equivalent calcitriol dose (e.g. 10 µg of maxacalcitol was equivalent to 1.5 µg of calcitriol). Both male and female patients were included in the present study.

## Efficacy and safety endpoints

The efficacy endpoints were 30 weeks of mean serum levels of intact PTH, cCa, phosphorus, and intact and C-terminal FGF23 levels, and their percent changes from baseline, as well as numbers and percentages of patients who achieved the targets of intact PTH (60 to 240 pg/mL), cCa (8.4 to 10.0 mg/dL), and phosphorus (3.5 to 6.0 mg/dL) during the evaluation period [35, 36], stratified by the VDRA dose. Ratios of intact-to-C-terminal FGF23 in all patients by VDRA dose, as well as the mean expression levels of bone turnover markers (bone specific alkaline phosphatase [BAP] and tartrate-resistant acid phosphatase-5b [TRACP-5b]) by VDRA dose were evaluated. The mean daily dose of evocalcet was presented by VDRA dose.

Safety endpoints were incidences of pre-determined hypocalcemia-related (cCa decreased, blood calcium decreased, and hypocalcemia) and upper gastrointestinal tract-related (nausea,

vomiting, abdominal discomfort, abdominal distension, and decreased appetite) adverse drug reactions (ADRs) by VDRA dose.

## Statistical methods

Efficacy was evaluated in the per protocol set (PPS), which excluded the following patients: those who did not receive treatment with evocalcet, who had no intact PTH measurement after starting the treatment with evocalcet, who did not meet any inclusion criteria or met any of the exclusion criterion, who were prescribed evocalcet for $\geq 28$ weeks with an adherence rate of $< 70\%$, who received prohibited concomitant medication or therapy, who did not have two or more intact PTH measurements from three timepoints during the evaluation period (Weeks 28 to 30), or who had a protocol violation that might have affected the efficacy evaluations. Safety was assessed in the safety analysis set, which included all patients except those who did not receive treatment with evocalcet.

For the patient baseline characteristics, categorical data were summarized using frequencies and percentages, and continuous data were summarized using descriptive statistics consisting of the number of patients, mean, and standard deviation by DG. Categorical data were analyzed using the $\chi^2$ test, and continuous data were analyzed using the Kruskal–Wallis test or analysis of variance according to the date distribution. For between-group comparisons of the percentages of patients who achieved the targets, crude differences and 95% confidence intervals were calculated, which were adjusted further for age ($< 65$ and $\geq 65$ years), sex, intact PTH level ($< 500$ pg/mL and $\geq 500$ pg/mL) at baseline and previous cinacalcet treatment. The $\chi^2$ test was used for statistical analysis. Differences in changes in intact PTH, corrected calcium, and phosphorus from baseline to Week 30 among three DGs were analyzed by a repeated linear mixed model. Pearson's correlation coefficient and linear mixed-effects modeling were used to assess the relationship between intact and C-terminal FGF23 levels, and the level of correlation was shown as adjusted R-squared. The ratio of intact-to-C-terminal FGF23 at each time point was normalized to the baseline ratio in each DG. For intact FGF23 levels, those $\geq 80,000$ pg/mL were excluded from the analyses. Changes in the ratio of intact-to-C-terminal FGF23 between Week 0 and Week 30 were analyzed by a generalized linear mixed model. All pre-determined ADRs were summarized by preferred term according to the Medical Dictionary for Regulatory Activities version 19.0, and between-group differences were analyzed using the $\chi^2$ test. Statistical analyses were performed using SAS version 9.4 (SAS Institute, Cary, NC, USA), and two-tailed $p$-values $< 0.05$ were considered statistically significant.

## Results

### Patients

Of 320 patients randomized to receive evocalcet in the previous phase 3 head-to-head comparison study [26], 317 patients were included in the safety analysis set and 253 patients were included in the PPS (117, 45, and 91 were treated with no, low, and high IV VDRA, respectively) in this *ad hoc* analysis.

Patient baseline characteristics were similar among the three DGs in the PPS, including the proportions of males and females (range: males 65.8% to 76.9% and females 23.1% to 34.2%) (Table 1). However, a significantly higher number of prior cinacalcet users were included in the HDG compared with the other DGs ($p < 0.001$), and the baseline mean BAP level was lower ($p = 0.015$) and the C-terminal FGF23 level was higher ($p = 0.031$) as the concomitant VDRA dose range increased.

**Table 1. Baseline characteristics of study patients.**

| Parameter (unit) | | Weekly VDRA dose | | | |
|---|---|---|---|---|---|
| | | 0 | < 1.5 µg | ≥ 1.5 µg | p-value[a] |
| | | N = 117 | N = 45 | N = 91 | |
| Male, n (%) | | 77 (65.8) | 30 (66.7) | 70 (76.9) | 0.193 |
| Age (years) | | 62.4 (10.6) | 62.0 (11.9) | 58.6 (11.8) | 0.050 |
| Body mass index (kg/m$^2$) | | 24.18 (4.23) | 25.36 (5.24) | 24.70 (4.60) | 0.317 |
| Primary disease, n (%) | Diabetic nephropathy | 35 (29.9) | 11 (24.4) | 20 (22.0) | 0.464 |
| | Chronic glomerulonephritis | 47 (40.2) | 15 (33.3) | 42 (46.2) | |
| | Nephrosclerosis | 14 (12.0) | 5 (11.1) | 12 (13.2) | |
| | Other | 21 (17.9) | 14 (31.1) | 17 (18.7) | |
| Duration of dialysis (months) | | 121.2 (86.7) | 122.0 (82.6) | 135.4 (87.4) | 0.468 |
| Dry weight (kg) | | 60.51 (12.97) | 64.02 (15.52) | 63.54 (14.10) | 0.188 |
| Dialysis efficiency (spKt/V) | | 1.510 (0.285) | 1.473 (0.299) | 1.488 (0.285) | 0.730 |
| Cinacalcet use before screening, n (%) | | 61 (52.1) | 24 (53.3) | 70 (76.9) | < 0.001 |
| VDRA use at Week 0, n (%) | | 81 (69.2) | 45 (100.0) | 91 (100.0) | < 0.0001 |
| | IV | 0 (0.0) | 45 (100.0) | 91 (100.0) | |
| | Oral | 81 (69.2) | 0 (0.0) | 0 (0.0) | - |
| Intact PTH (pg/mL) | | 438.9 (197.7) | 382.2 (125.8) | 409.6 (168.0) | 0.183 |
| Corrected calcium (mg/dL) | | 9.5 (0.6) | 9.5 (0.5) | 9.6 (0.5) | 0.374 |
| Phosphorus (mg/dL) | | 5.6 (1.2) | 5.7 (1.4) | 5.9 (1.4) | 0.383 |
| Calcium corrected-phosphate product (mg$^2$/dL$^2$) | | 53.35 (11.00) | 54.46 (13.48) | 56.36 (13.23) | 0.217 |
| BAP (µg/L) | | 18.79 (10.63) | 18.42 (10.66) | 15.61 (8.18) | 0.015 |
| TRACP-5b (mU/dL) | | 816.00 (387.62) | 773.51 (392.01) | 745.76 (405.17) | 0.184 |
| Total P1NP (µg/L) | | 452.51 (304.75) | 461.33 (297.93) | 405.86 (258.40) | 0.530 |
| Whole PTH (pg/mL) | | 203.58 (91.73) | 190.63 (79.76) | 190.48 (89.93) | 0.274 |
| Intact FGF23 (pg/mL) | | 18,082.25 (19,267.33) | 21,363.44 (20,641.90) | 25,365.10 (24,136.73) | 0.051 |
| C-terminal FGF23 (RU/mL) | | 14,414.73 (17,677.51) | 18,293.33 (23,928.71) | 23,488.03 (34,396.35) | 0.031 |
| Largest parathyroid gland volume (mm$^3$) | | 321.97 (566.19) | 256.92 (261.39) | 364.11 (791.89) | 0.846 |

Unless otherwise specified, data are presented as the mean (standard deviation).

[a]$\chi^2$ test (categorical), Kruskal–Wallis test or analysis of variance (continuous).

Abbreviations: BAP, bone specific alkaline phosphatase; FGF23, fibroblast growth factor-23; IV, intravenous; PTH, parathyroid hormone; P1NP, procollagen type 1 amino-terminal propeptide; TRACP-5b, tartrate-resistant acid phosphatase-5b; VDRA, vitamin D receptor activator.

### Intact PTH, cCa, phosphorus, and bone turnover markers

After the initiation of treatment with evocalcet, the mean intact PTH level gradually decreased in all DGs by Week 30 (NDG vs LDG, $p > 0.1$; NDG vs HDG, $p < 0.0001$; LDG vs HDG, $p = 0.0025$). The mean percent reductions in intact PTH from baseline were also increased throughout the 30 weeks of treatment, irrespective of concomitant VDRA dose (Fig 1A, S2 Table). The mean cCa level decreased until Week 4 after treatment initiation and remained at a similar level in all DGs with slightly lower levels in the NDG and LDG compared with the HDG (NDG vs LDG, $p > 0.1$; NDG vs HDG, $p = 0.0019$; LDG vs HDG, $p > 0.1$). A similar tendency was observed for the mean percent changes in cCa levels, irrespective of concomitant VDRA dose (Fig 1B, S2 Table). The mean phosphorus levels and mean changes in the phosphorus levels tended to show moderate reductions over the study period (all between-groups, $p < 0.0001$) (Fig 1C, S2 Table). However, no significant difference was observed for the intact PTH, cCa, or phosphorus level among the three DGs when the levels were adjusted according to their respective baseline levels (all cases, $p > 0.95$).

The mean BAP level was slightly increased at Week 6 and gradually returned to the baseline level at Week 30 in all DGs (Fig 2A). Mean TRACP-5b levels gradually decreased in all DGs throughout the 30 weeks of the study period (Fig 2B).

### Intact PTH, cCa, and phosphorus guideline target achievement rates

The proportions of patients who achieved the intact PTH target in the LDG and HDG were slightly higher than that in the NDG, but no significant difference was found among the DGs.

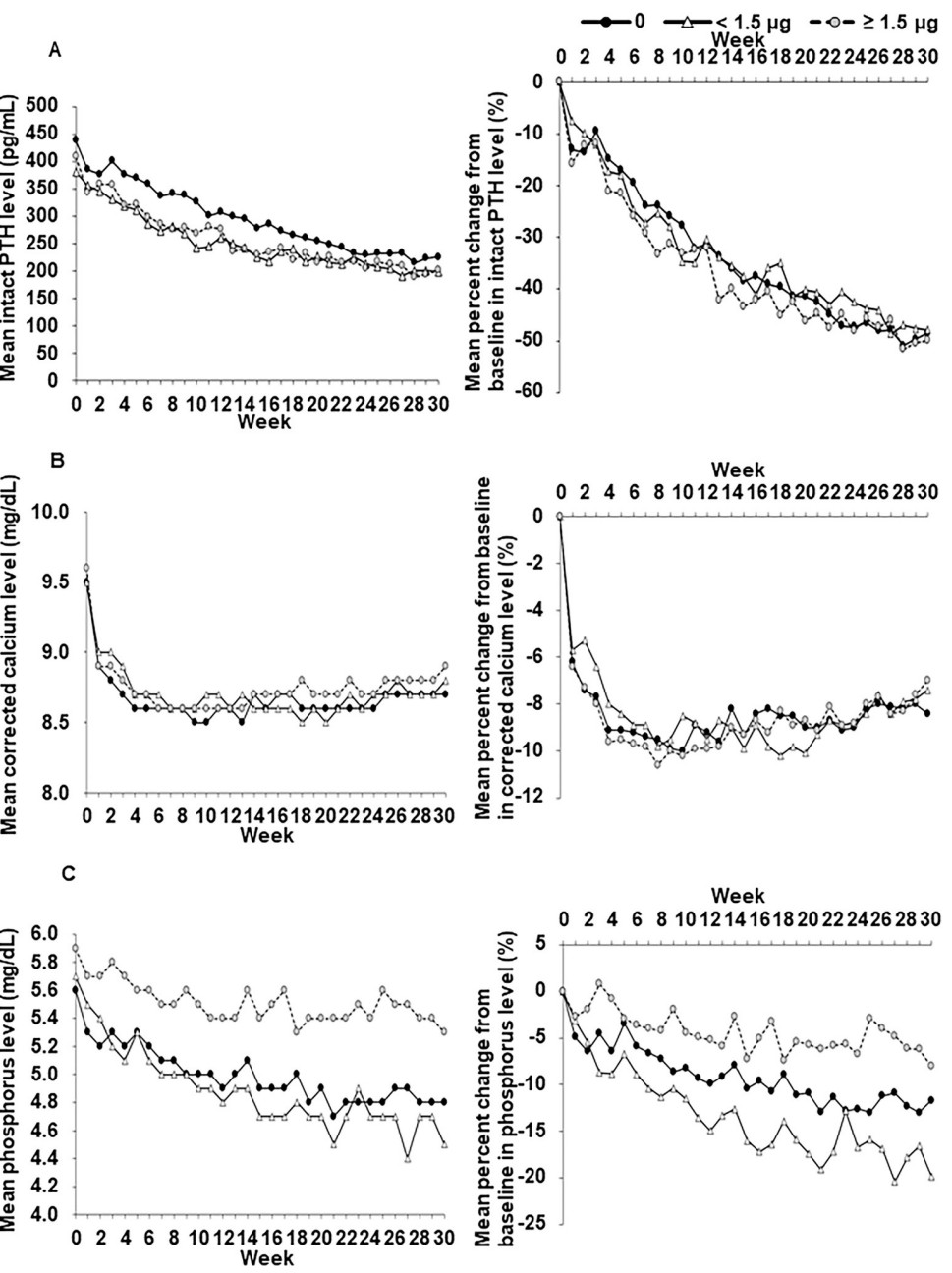

**Fig 1.** Mean levels (left) and mean percent changes from baseline (right) in (A) intact PTH, (B) corrected calcium, (C) and phosphorus levels stratified by concomitant baseline IV VDRA weekly dose in patients with SHPT treated with evocalcet for 30 weeks. IV, intravenous; PTH, parathyroid hormone; SHPT, secondary hyperparathyroidism; VDRA, vitamin D receptor activator.

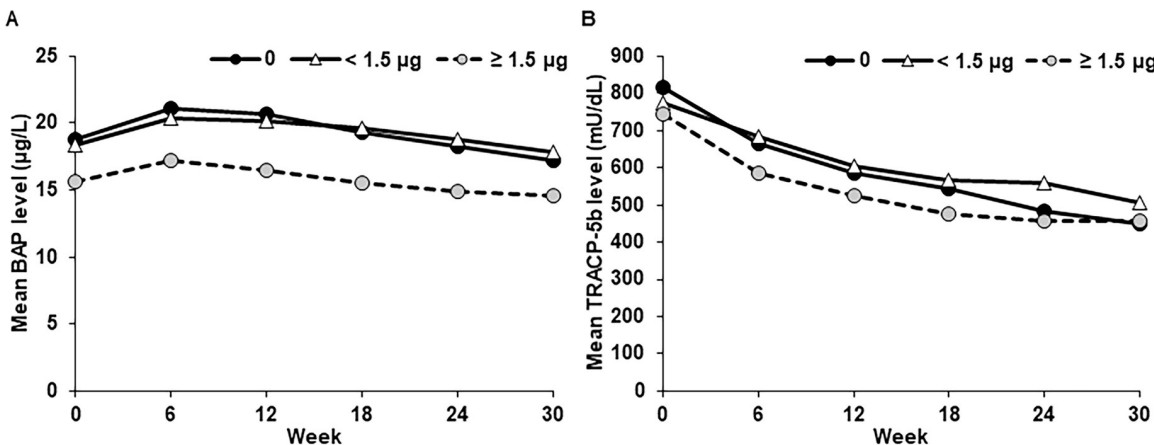

**Fig 2.** Mean levels of serum BAP (A) and TRACP-5b (B), stratified by concomitant baseline IV VDRA. BAP, bone specific alkaline phosphatase; IV, intravenous; SHPT, secondary hyperparathyroidism; TRACP-5b, tartrate-resistant acid phosphatase-5b; VDRA, vitamin D receptor activator.

The proportion of patients who achieved the cCa target was significantly higher as the VDRA dose increased ($p$ = 0.043). The proportion of patients who achieved the phosphorus target was similar among the DGs. The proportion of patients who achieved intact PTH, cCa and phosphorus targets was similar among the DGs, although it was slightly higher in the HDG compared with the other DGs (Table 2).

## Daily evocalcet doses

The mean daily evocalcet doses were increased at Week 4, when a dose increase was permitted, and showed a gradual increase until the end of the dose adjustment period in all DGs (mean ± standard deviation at Week 27: 3.32 ± 2.38, 3.91 ± 2.46, and 3.34 ± 2.43 mg/day in the

**Table 2. Numbers, percentages, and between-group differences of patients who achieved the target parameters during Weeks 28 to 30 stratified by concomitant baseline IV VDRA dose.**

| | Weekly VDRA dose | | | |
|---|---|---|---|---|
| | **0** | **< 1.5 μg** | **≥ 1.5 μg** | **$p$-value[a]** |
| **Patients who achieved the guideline target level** | **N = 117** | **N = 45** | **N = 91** | |
| Intact PTH, n (%) | 80 (68.4) | 34 (75.6) | 70 (76.9) | 0.349 |
| Difference (95% CI) | | 7.2 (−9.01, 20.72) | 8.5 (−3.81, 20.16) | |
| Adjusted difference (95% CI) | | 4.6 (−11.90, 18.50) | 11.5 (−1.83, 23.83) | |
| Corrected calcium, n (%) | 70 (59.8) | 32 (71.1) | 69 (75.8) | 0.043 |
| Difference (95% CI) | | 11.3 (−5.47, 25.66) | 16.0 (3.12, 27.84) | |
| Adjusted difference (95% CI) | | 9.8 (−7.35, 24.91) | 16.4 (2.75, 28.86) | |
| Phosphorus, n (%) | 86 (73.5) | 31 (68.9) | 58 (63.7) | 0.318 |
| Difference (95% CI) | | −4.6 (−20.83, 9.84) | −9.8 (−22.27, 2.81) | |
| Adjusted difference (95% CI) | | −4.6 (−21.20, 10.31) | −13.2 (−26.23, 0.17) | |
| The above three parameters, n (%) | 45 (38.5) | 16 (35.6) | 38 (41.8) | 0.768 |
| Difference (95% CI) | | −2.9 (−18.21, 13.90) | 3.3 (−9.89, 16.51) | |
| Adjusted difference (95% CI) | | −5.1 (−21.00, 12.19) | 4.2 (−9.94, 18.21) | |

[a]$\chi^2$ test.

Adjusted for baseline age, sex, intact PTH level, and previous cinacalcet treatment.

Abbreviations: CI, confidence interval; IV, intravenous; PTH, parathyroid hormone; VDRA, vitamin D receptor activator.

**Table 3. Incidences of ADRs stratified by concomitant baseline IV VDRA dose.**

| | Weekly VDRA dose | | | All | |
| --- | --- | --- | --- | --- | --- |
| | 0 | < 1.5 µg | ≥ 1.5 µg | | p-value[a] |
| | N = 152 | N = 55 | N = 110 | N = 317 | |
| Hypocalcemia-related ADRs, n (%) | 40 (26.3) | 5 (9.1) | 14 (12.7) | 59 (18.6) | 0.003 |
| Corrected calcium decreased | 23 (15.1) | 2 (3.6) | 12 (10.9) | 37 (11.7) | 0.072 |
| Blood calcium decreased | 7 (4.6) | 3 (5.5) | 1 (0.9) | 11 (3.5) | 0.184 |
| Hypocalcemia | 10 (6.6) | 0 (0.0) | 1 (0.9) | 11 (3.5) | 0.014 |
| Upper gastrointestinal tract-related ADRs, n (%) | 16 (10.5) | 10 (18.2) | 15 (13.6) | 41 (12.9) | 0.337 |
| Nausea | 10 (6.6) | 1 (1.8) | 5 (4.5) | 16 (5.0) | 0.368 |
| Vomiting | 6 (3.9) | 4 (7.3) | 4 (3.6) | 14 (4.4) | 0.522 |
| Abdominal discomfort | 4 (2.6) | 1 (1.8) | 5 (4.5) | 10 (3.2) | 0.562 |
| Abdominal distension | 0 (0.0) | 1 (1.8) | 1 (0.9) | 2 (0.6) | 0.311 |
| Decreased appetite | 3 (2.0) | 3 (5.5) | 2 (1.8) | 8 (2.5) | 0.312 |

[a]$\chi^2$ test.

Abbreviation: ADR, adverse drug reaction; IV, intravenous; VDRA, vitamin D receptor activator.

NDG, LDG, and HDG, respectively) with a slightly higher mean dose in the LDG compared with the other DGs.

## ADRs

Overall, 18.6% (59/317) of patients experienced pre-determined hypocalcemia-related ADRs, and a significant difference was found among the three DGs with the highest incidence rate in the NDG ($p = 0.003$). The incidence of hypocalcemia in the NDG was the highest compared with the LDG and HDG ($p = 0.014$). Also, 12.9% (41/317) of patients experienced pre-determined upper gastrointestinal tract-related ADRs without significant differences in the incidence rates (Table 3).

## FGF23

The mean intact and C-terminal FGF23 levels in the HDG were higher than those in the NDG and LDG at baseline and throughout the treatment period. After starting treatment with evocalcet, the intact and C-terminal FGF23 levels were decreased in all DGs (Fig 3). The percent reductions from baseline in intact FGF23 and C-terminal FG23 in the LDG were significantly higher than those in the HDG throughout the 30 weeks of treatment ($p < 0.005$ and $p < 0.030$, respectively). The ratio of intact-to-C-terminal FGF23 in all DGs was 1.2 from Week 0 to the measurement time point at Week 6, which indicates a higher intact FGF23 level compared with the C-terminal FGF23 level (Fig 4). The ratio in the NDG and LDG decreased to 1.1 at Week 12 and then to 0.9 at Week 30 (vs HDG, $p < 0.007$); however, the reduction in the HDG was not as marked (1.1 at Week 30).

The adjusted R-squared was similar among the DGs in Week 0 and did not show a notable change at Week 30 (Fig 5). The slope of the FGF23 intact and C-terminal relationship at Week 0 and Week 30 was 0.6717 (95% CI, 0.6489, 0.6946) and 0.7230 (95% CI, 0.7038, 0.7422) in all patients, 0.6359 (95% CI, 0.5981, 0.6736) and 0.6927 (95% CI, 0.6583, 0.7271) in the NDG, 0.6596 (95% CI, 0.6228, 0.6965) and 0.6950 (95% CI, 0.6436, 0.7464) in the LDG, and 0.6936 (95% CI, 0.6566, 0.7307) and 0.7376 (95% CI, 0.7085, 0.7667) in the HDG, respectively. The slope of the FGF23 intact and C-terminal relationship significantly increased after treatment with evocalcet at Week 30 compared with Week 0 in all patients (0.04874 [standard error 0.01307], $p = 0.0002$) and patients in the NDG (0.06729 [standard error 0.01917], $p = 0.0005$).

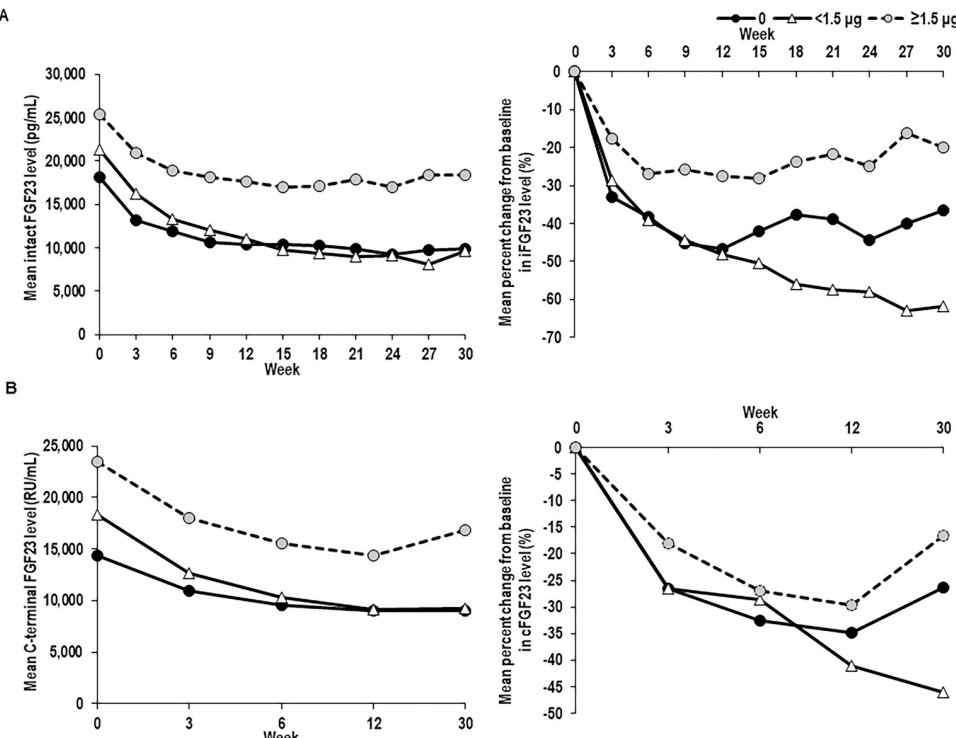

**Fig 3.** Mean levels (left) and mean percent changes from baseline (right) in (A) intact FGF23 and (B) C-terminal FGF23 levels stratified by concomitant baseline IV VDRA weekly dose in patients with SHPT treated with evocalcet for 30 weeks. IV, intravenous; SHPT, secondary hyperparathyroidism; VDRA, vitamin D receptor activator. FGF23, fibroblast growth factor-23.

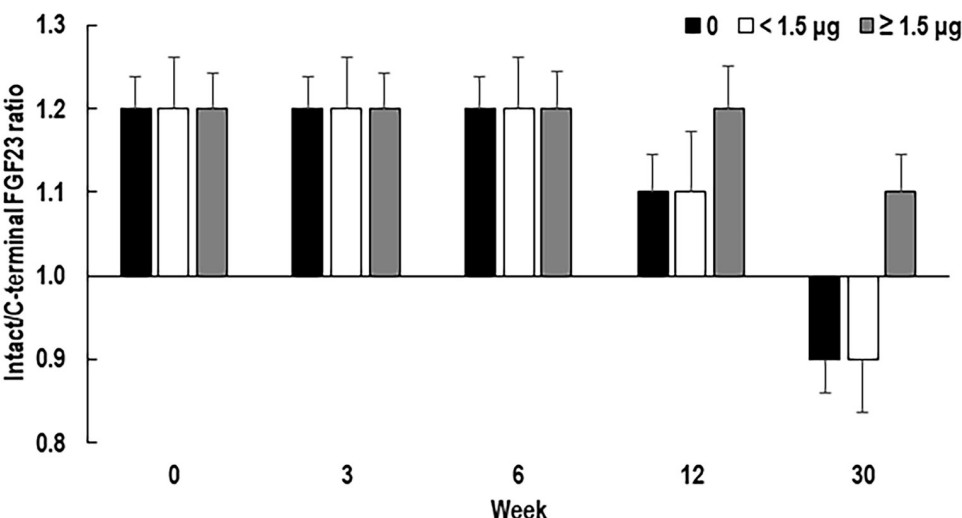

**Fig 4. Intact FGF23-to-C-terminal FGF23 mean (standard error) ratios, stratified by concomitant baseline IV VDRA.** FGF23, fibroblast growth factor-23; IV, intravenous; SHPT, secondary hyperparathyroidism; VDRA, vitamin D receptor activator.

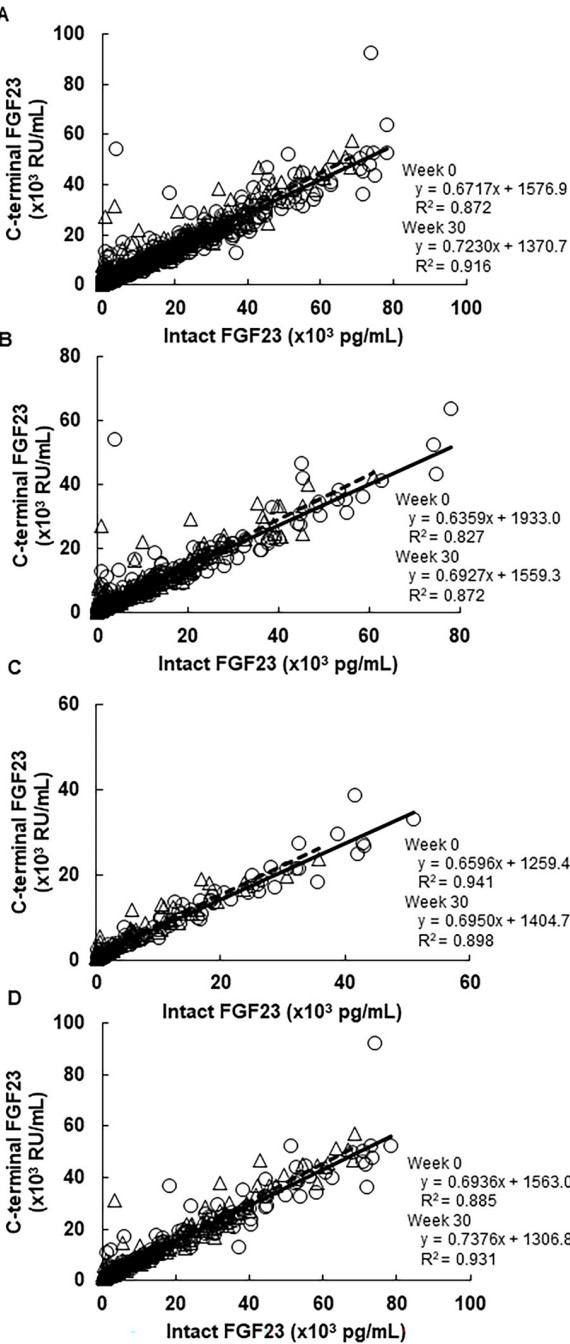

**Fig 5. Correlations between intact FGF23 and C-terminal FGF23 at Week 0 and Week 30.** (A) all patients, (B), patients with no concomitant treatment of IV VDRA, (C) patients with IV VDRA < 1.5 μg/week, and (D) patients with IV VDRA ≥ 1.5 μg/week. Circles with a sold line, Week 0; triangles with a broken line, Week 30. FGF23, fibroblast growth factor-23; IV, intravenous; VDRA, vitamin D receptor activator.

## Discussion

The current *ad hoc* analyses demonstrated that evocalcet was effective at reducing the intact PTH, cCa, phosphorus, TRACP-5b, BAP, and FGF23 levels, irrespective of concomitant IV VDRA. The clinical benefit of evocalcet with IV VDRA was also found for hypocalcemia.

Although the difference was not statistically significant, we found that over the 30 weeks of treatment, intact PTH levels tended to be slightly lower in the LDG and HDG compared with the NDG; indeed, > 7% of patients achieved the target with VDRA compared with those receiving evocalcet alone, which indicates that evocalcet and VDRA might have additive effects. Treatment with evocalcet and VDRA showed similar but significant effects on the cCa levels: > 10% patients achieved the target compared with evocalcet alone. Treatment with evocalcet alone decreased cCa levels below the target range in some patients, which was likely prevented by concomitant VDRA, thereby improving the achievement rate. Indeed, patients treated with concomitant VDRA developed hypocalcemia-related ADRs less frequently than those without VDRA. Therefore, evocalcet and VDRA appear to complement each other in the regulation of calcium levels, leading to further clinical improvements. However, unlike intact PTH and cCa, VDRA had no impact on the proportion of patients who achieved the phosphorus target.

Accumulating evidence suggests that elevated FGF23 levels are associated with increased mortality in patients undergoing hemodialysis [7–10]. Although the mechanism by which elevated circulating FGF23 levels increase mortality is not fully understood, it is accepted that FGF23 levels should be better controlled [15]. As previously reported for cinacalcet [12], our results showed that evocalcet decreased intact and C-terminal FGF23 levels. However, overall serum FGF23 levels tended to be higher as the VDRA dose increased, consistent with previous studies [30–32]. Furthermore, while the percent changes in the HDG stopped decreasing after Week 12, those in the LDG kept decreasing, which suggests that a combination of low range IV VDRA and evocalcet provides better control over FGF23 levels.

A lowered intact-to-C-terminal FGF23 ratio indicates a reduction in FGF23 synthesis; therefore, our results suggest that evocalcet decreased FGF23 levels by suppressing FGF23 synthesis irrespective of concomitant VDRA, although this effect was significantly lower in the HDG compared with the other DGs partly because VDRA stimulates FGF23 synthesis. The suppression of FGF23 synthesis by evocalcet was further supported by our finding that the slope at Week 0 was increased at Week 30 with the highest slope change found in the NDG. In addition to FGF23, favorable effects of the combination therapy were found for levels of the bone turnover markers TRACP-5b and BAP. The timing of the changes in the FGF23 coincided with that of BAP. Because FGF23 is synthesized in osteoblasts and osteocytes [5, 6], it would be of clinical interest to elucidate any interactions among these markers and their clinical implications on the prognosis of SHPT in responding to a combination therapy with calcimimetics and VDRA. Furthermore, because elevated levels of FGF23 and bone turnover markers have been presented as prognostic factors [7, 12, 37, 38], further studies are needed to evaluate the long-term effects of this combination therapy.

We observed the clinical benefit of combination therapy with evocalcet and VDRA for predetermined hypocalcemia-related ADRs. Although upper gastrointestinal tract-related ADRs were slightly increased by the addition of VDRA, this increase was negligible, and no notable new hypocalcemia- or upper gastrointestinal tract-related safety concerns were observed.

In this study, the daily evocalcet dose in the LDG was slightly higher than that in the other DGs. Because evocalcet and VDRA both reduce serum intact PTH levels, patients receiving the lower VDRA dose range were assumed to be treated with a higher range of evocalcet daily doses to supplement the efficacy of VDRA. The combination of IV VDRA and evocalcet dose ranges applied to patients in the LDG appeared to be more beneficial than in the HDG because LDG patients had equivalent efficacy to the HDG but with potentially better long-term prognosis related to lower phosphorus and FGF23 levels. However, differences among the three DGs obtained in this study, such as an elevated FGF23 level [29–34], and/or differential patient baseline characteristics that included a higher proportion of patients with prior cinacalcet use

in the HDG, might have been a consequence of the ongoing VDRA treatment. A *post hoc* analysis of the previously conducted Evaluation of Cinacalcet Therapy to Lower Cardiovascular Events (EVOLVE) trial, in which the effects of cinacalcet on cardiovascular disease were investigated in severe SHPT patients on maintenance hemodialysis treatment, reported that, in most patients, the doses of cinacalcet, VDRA, and phosphate binders were not adjusted just after or at $\geq$ 14 days after the onset of hypocalcemia [39], which indicates that further studies are needed to confirm whether a high dose range of VDRA is beneficial for the treatment of SHPT patients with oral calcimimetics. Considering the advantages and disadvantages of the combination therapy, treatment options and doses need to be finely adjusted dependent on individual patient disease state, conditions, and characteristics.

This study had some limitations. First, differences in the 30 weeks of mean changes and percent changes from baseline in the intact and C-terminal FGF23, BAP, and TRACP-5b levels were not statistically tested within and among the three DGs. Therefore, any longitudinal mean changes and between group differences in levels need to be interpreted as tendencies. In addition, all patients were Japanese, the study had a small sample size, and patients were treated with evocalcet only up to 30 weeks. Although evocalcet was developed in Japan and has only recently been approved for the treatment of SHPT in patients on maintenance dialysis, it is expected that long-term studies will be conducted in the near future to determine the efficacy and safety of evocalcet in other global populations with larger sample sizes.

In conclusion, treatment with evocalcet improved the achievement rate of the serum cCa target, decreased FGF23 levels by suppressing FGF23 synthesis, and decreased the incidence of hypocalcemia in the presence of VDRA, demonstrating the additive benefits of the combination therapy in SHPT patients.

## Supporting information

**S1 Table. List of Institutional Review Boards (IRBs).**
(DOCX)

**S2 Table. Standard deviations of the mean levels and mean percent changes from baseline in intact PTH, corrected calcium, and phosphorus levels stratified by concomitant baseline IV VDRA weekly dose in patients with SHPT treated with evocalcet for 30 weeks.**
(DOCX)

## Acknowledgments

The authors thank all collaborators for their participation in the study, Ms. Shiori Yasui, Mr. Taichi Mizogui, and Dr. Shinjo Yada of A2 Healthcare Corporation for statistical analysis, and ASCA Corporation for medical writing assistance.

## Author Contributions

**Conceptualization:** Takashi Shigematsu, Masafumi Fukagawa, Tadao Akizawa.

**Data curation:** Shinji Asada, Yuichi Endo, Takehisa Kawata.

**Formal analysis:** Shinji Asada, Yuichi Endo, Takehisa Kawata.

**Investigation:** Shinji Asada, Yuichi Endo, Takehisa Kawata.

**Methodology:** Shinji Asada, Yuichi Endo, Takehisa Kawata.

**Writing – original draft:** Shinji Asada, Takehisa Kawata.

**Writing – review & editing:** Takashi Shigematsu, Masafumi Fukagawa, Tadao Akizawa.

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
