## [Decision Letter · Decision Letter 0]

13 Apr 2021

PONE-D-21-07286

Evocalcet with vitamin D receptor activator treatment for secondary hyperparathyroidism

PLOS ONE

Dear Dr. SHIGEMATSU

Thank you for submitting your manuscript to PLOS ONE. After careful consideration, we feel that it has merit but does not fully meet PLOS ONE’s publication criteria as it currently stands. Therefore, we invite you to submit a revised version of the manuscript that addresses the points raised during the review process.

We look forward to receiving your revised manuscript.

Kind regards,

Pasqual Barretti, Ph.D., MD

Academic Editor

PLOS ONE

Journal Requirements:

'TS received consulting fees from Kyowa Kirin Co., Ltd. (KKC), Ono Pharmaceutical,

Taisho Toyama Pharmaceutical, Fuji Pharma, and FUSO, and lecture fees from KKC,

Chugai Pharmaceutical, Bayer, Kissei Pharmaceutical, Torii Pharmaceutical, Ono

Pharmaceutical, and FUSO. SA, YE, and TK are employees of KKC. MF received

consulting fees from KKC and Ono Pharmaceutical; lecture fees from KKC, Bayer, Torii

Pharmaceutical, and Ono Pharmaceutical; and grants from KKC and Bayer. TA

received consulting fees from KKC, Astellas Pharma, Bayer, Fuso Pharmaceutical,

Japan Tobacco, Ono Pharmaceutical, Sanwa Chemical, Otsuka, GSK, and NIPRO,

and lecture fees from KKC, Chugai Pharmaceutical, Bayer, Kissei Pharmaceutical,

Torii Pharmaceutical, and Ono Pharmaceutical. The authors report no other conflicts of

interest in this work.'

Additional Editor Comments (if provided):

The reviewers, in particular the reviewers 1 and 2 have made important questions and suggestions regrading approach, lack of some information, methodological e statistical aspect , which have be considered and addressed by the authors. Therefore, a deep revision is still necessary at this moment.

Reviewers' comments:

Reviewer's Responses to Questions

**Comments to the Author**

1. Is the manuscript technically sound, and do the data support the conclusions?

Reviewer #1: Partly

Reviewer #2: Yes

Reviewer #3: Partly

2. Has the statistical analysis been performed appropriately and rigorously? 

Reviewer #1: No

Reviewer #2: Yes

Reviewer #3: No

3. Have the authors made all data underlying the findings in their manuscript fully available?

Reviewer #1: No

Reviewer #2: Yes

Reviewer #3: Yes

4. Is the manuscript presented in an intelligible fashion and written in standard English?

Reviewer #1: Yes

Reviewer #2: Yes

Reviewer #3: Yes

5. Review Comments to the Author

Reviewer #1: Major comments

1. If I understand correctly the present analysis has been performed only in the patients of the evocalcet group of the parent head-to-head comparison study. This should be made clear from the outset. In the parent trial 320 patients were randomized to receive evocalcet. In the present ad hoc analysis only 253 patients were included. Why were the remaining 67 patients excluded ?

2. VDRA stands for calcitriol and a group of other active vitamin D compounds. Authors further need to tell from the outset which active vitamin D compounds were administered, and to which extent the administered doses were comparable.

3. Eighty-one patients among the 117 weekly zero VDRA dose patients were on oral VDRA therapy at baseline. Did they continue or was this therapy stopped at trial initiation ?

4. In the Results section authors repeatedly mention « trends » for changes of various biochemistry values but actually fail to provide P values in the Figures or Figure legends. Therefore, I conclude that none of these « trends » corresponded to changes reaching the level of statistical significance. The terms « trend » and « trended » should be deleted and instead any non significant change should be acknowledged as no change. This will require a serrious revision of the interpretation of numerous results.

5. Strictly speaking, hypocalcemia is not an ADR, all the more since the calcimimetic-induced decrease of serum PTH goes along with a decrease in serum calcium, at least in ESKD patients on dialysis therapy. It would be interesting to have information on usual hypocalcemia-associated clinical symptoms and signs.

6. In a related post-hoc analysis of the EVOLVE trial in hemodialysis patients receiving cinacalcet therapy - not quoted by the authors - Floege et al found that in the majority of hypocalcemia episodes changes in calcium administration were the preferred treatment option (DOI: 10.1016/j.kint.2017.12.014). In that trial, adjustments of cinacalcet or vitamin D sterol doses were made only in a minority of hypocalcemia episodes. This observation, together with the observation in the present study that high-dose VDRA treatment was associated with significantly higher serum FGF23 levels than low-dose VDRA treatment, and the notion that higher FGF23 levels are associated with higher mortality risk, would rather plead against systematic VDRA administration in dialysis patients on calcimimetic therapy. The authors may want to make this clear in the Discussion.

Minor remarks

Abstract should tell that evocalcet was administered orally. Moreover, authors need to indicate, both in Abstract and in Methods section, that the head-to-head comparison of the parent trial was between evocalcet and cinacalcet.

Abstract L37. I wonder whether the term « clinical benefits » is appropriate here, at least as far as FGF23 is concerned. As to the lesser incidence of hypocalcemia I would like to know whether concomitant vitamin D receptor activator treatment was more effective in avoiding hypocalcemia-associated symptoms and signs.

Introduction. The statement that IV administration of VDRA « is the

standard therapy for patients with SHPT » is incorrect as such. At least in the Western world VDRA can by given either orally or intravenously for the treatment of SHPT.

Discussion, lines 238-284. The statement is inappropriate that « intact PTH levels were slightly lower in the LDG and HDG compared with the NDG » since the difference was not statistically significant.

Reviewer #2: This study is an ad hoc analysis of a previous phase 3, head-to-head comparison study of hemodialysis patients with hyperparathyroidism, after 30-day treatment with evocalcet. This study was conducted at 89 study sites in Japan (Oct 2015-Nov 2016) and demonstrated a non-inferiority to cinacalcet to suppress iPTH with fewer gastro-intestinal adverse events.

This study has demonstrated that the association of oral evocalcet with vitamin D activator was beneficial, with lower synthesis of FGF-23, lower incidence of hypocalcemia and with more patients achieving the calcium target. This oral formulation of evocalcet, was also associated with a reduction of gastric symptoms when compared with oral cinacalcet.

This study confirms the benefit of the treatment of hyperparathyroidism with an association of a calcimimetic + vitamin D.

The reduction of gastric symptoms verified with evocalcet will contribute for a better adherence of this new oral treatment for hyperparathyroidism.

However, the intravenous administration of etelcalcetide after hemodialysis, assures the patient adherence to this kind of treatment.

Evocalcet will be very useful for the treatment of patients with hyperparathyroidism who develop gastric symptoms associated with cinacalcet and who have no conditions for a three times/week intravenous treatment. This will be the case, for instance, of DP patients.

Reviewer #3: This paper reported an ad-hoc analysis results using patients enrolled in one treatment arm of a completed phase III trial. Patients were categorized into three groups according to average weekly VDRA dose. Many clinical outcomes and adverse drug reactions were compared among these three groups. However, almost all statistical analysis only employed basic statistical tests and multiple testing adjustment was not implemented. In addition, most statements about the dose effect of VDRA were without rigorous statistical evidences presented. A professional statistician is recommended to be involved for rigorous data analysis and drawing conclusions.

Specifically,

1. Figure 1 displayed longitudinal mean measurements for each group of patients. It would be helpful to also include variations in the plots. In addition, linear mixed models are recommended to do a formal comparison among three patients groups and make conclusions rather than just looking at the plots and stating there was a tendency of VDRA dose-dependent trend.

2. Table 1 implies that these three groups of patients were not balanced in all baseline measurements. Therefore, the corresponding result comparisons should consider adjusting for these possible confounding factors.

3. A conclusion from Figure 4 says "The slope of the FGF23 intact and C-terminal relationship increased after treatment with evocalcet at Week 30 compared with Week 0 in all patients, as well as by VDRA DG." First, the estimated slope were not reported. Second, a formal statistical analysis should be implemented here to provide support of such statement.

Minor comments:

1. The title of Table 2 says "Mean numbers and percentages...". However, outcomes in Table 2 are all binary ones. It is confusing why there would be mean numbers and percentages.

2. It is also helpful to add in variation of those ratios to Figure 3.

6. PLOS authors have the option to publish the peer review history of their article (what does this mean?). If published, this will include your full peer review and any attached files.

Reviewer #1: No

Reviewer #2: No

Reviewer #3: No

---

## [Author Response · Author response to Decision Letter 0]

21 May 2021

Journal Requirements:

Response: File names have been corrected accordingly. 

Response: These data are now included as Fig. 3. 

3. Thank you for stating the following in the Competing Interests section: 'TS received consulting fees from Kyowa Kirin Co., Ltd. (KKC), Ono Pharmaceutical, Taisho Toyama Pharmaceutical, Fuji Pharma, and FUSO, and lecture fees from KKC, Chugai Pharmaceutical, Bayer, Kissei Pharmaceutical, Torii Pharmaceutical, Ono Pharmaceutical, and FUSO. SA, YE, and TK are employees of KKC. MF received consulting fees from KKC and Ono Pharmaceutical; lecture fees from KKC, Bayer, Torii Pharmaceutical, and Ono Pharmaceutical; and grants from KKC and Bayer. TA received consulting fees from KKC, Astellas Pharma, Bayer, Fuso Pharmaceutical, Japan Tobacco, Ono Pharmaceutical, Sanwa Chemical, Otsuka, GSK, and NIPRO, and lecture fees from KKC, Chugai Pharmaceutical, Bayer, Kissei Pharmaceutical, Torii Pharmaceutical, and Ono Pharmaceutical. The authors report no other conflicts of interest in this work.'

Response: As instructed, the updated Competing Interests Statement has been included in the cover letter. 

Reviewer #1: Major comments

1. If I understand correctly the present analysis has been performed only in the patients of the evocalcet group of the parent head-to-head comparison study. This should be made clear from the outset. In the parent trial 320 patients were randomized to receive evocalcet. In the present ad hoc analysis only 253 patients were included. Why were the remaining 67 patients excluded ?

Response: In the previous phase 3 head-to-head comparison study [Fukagawa et al Kidney Int. 2018;94: 818–825] and this study (Line 149), the efficacy of evocalcet was evaluated in the per protocol set as primary analysis, which was defined as the population of patients included in the full analysis set, excluding the following: those who failed to meet any of the inclusion criteria, or patients who met any of the exclusion criteria; those with a drug compliance of <70% (based on patient reports to investigators at hospital visits); those who received a prohibited concomitant medication or therapy; those with missing iPTH levels at more than one of three time points in the evaluation period; and those with a protocol deviation that may have affected the efficacy evaluation. According to the previous phase 3 head-to-head comparison study, three patients discontinued the study before initiating the treatment and, therefore, 317 patients started the treatment with evocalcet. Of these, 253 were included in the per-protocol set. This information was included in the original article of the phase 3 head-to-head comparison study. To clarify the data source, the citation has been added in Line 177. 

2. VDRA stands for calcitriol and a group of other active vitamin D compounds. Authors further need to tell from the outset which active vitamin D compounds were administered, and to which extent the administered doses were comparable.

Response: Specific active vitamin D compounds have been included in Line 112. A conversion method to show how their weekly doses were estimated has been included in Line 128. 

3. Eighty-one patients among the 117 weekly zero VDRA dose patients were on oral VDRA therapy at baseline. Did they continue or was this therapy stopped at trial initiation ?

Response: As stated in Line 110, initiation of or changes in preparation, dose, and dosing regimen of VDRA medications or derivatives were not permitted from 2 weeks before screening until the end of the study in all patients, meaning that patients in the zero VDRA group continued the same oral VDRA therapy. The sentence in Line 111 has been changed to clarify that the above point applies to the IV and oral VDRA therapies. 

4. In the Results section authors repeatedly mention « trends » for changes of various biochemistry values but actually fail to provide P values in the Figures or Figure legends. Therefore, I conclude that none of these « trends » corresponded to changes reaching the level of statistical significance. The terms « trend » and « trended » should be deleted and instead any non significant change should be acknowledged as no change. This will require a serrious revision of the interpretation of numerous results.

Response: The usage of “trend” and “dose-dependent difference” has been deleted. In addition, the lack of statistical analysis has been added as a limitation in Line 362.

5. Strictly speaking, hypocalcemia is not an ADR, all the more since the calcimimetic-induced decrease of serum PTH goes along with a decrease in serum calcium, at least in ESKD patients on dialysis therapy. It would be interesting to have information on usual hypocalcemia-associated clinical symptoms and signs.

Response: Although we understand the review comment, this was an ad hoc analysis of previously fixed data obtained in the head-to-head comparison study. As the reviewer pointed out, because of the clinical interest and significance of this study, future studies are needed to investigate the typical hypocalcemia-associated clinical symptoms and signs. 

6. In a related post-hoc analysis of the EVOLVE trial in hemodialysis patients receiving cinacalcet therapy - not quoted by the authors - Floege et al found that in the majority of hypocalcemia episodes changes in calcium administration were the preferred treatment option (DOI: 10.1016/j.kint.2017.12.014). In that trial, adjustments of cinacalcet or vitamin D sterol doses were made only in a minority of hypocalcemia episodes. This observation, together with the observation in the present study that high-dose VDRA treatment was associated with significantly higher serum FGF23 levels than low-dose VDRA treatment, and the notion that higher FGF23 levels are associated with higher mortality risk, would rather plead against systematic VDRA administration in dialysis patients on calcimimetic therapy. The authors may want to make this clear in the Discussion.

Response: Accordingly, the discussion has been changed in Line 353, and the post-hoc analysis of the EVOLVE trial by Floege et al. has been added to the References section. 

Minor remarks

Abstract should tell that evocalcet was administered orally. Moreover, authors need to indicate, both in Abstract and in Methods section, that the head-to-head comparison of the parent trial was between evocalcet and cinacalcet.

Response: The Abstract has been corrected by adding “oral” in Line 26. The Abstract and Materials and Methods sections have been corrected to clearly state that the previous phase 3 head-to-head comparison study investigated evocalcet and cinacalcet in Lines 24 and 89. 

Abstract L37. I wonder whether the term « clinical benefits » is appropriate here, at least as far as FGF23 is concerned. As to the lesser incidence of hypocalcemia I would like to know whether concomitant vitamin D receptor activator treatment was more effective in avoiding hypocalcemia-associated symptoms and signs.

Response: “Clinical benefits” has been changed to “benefits” in Lines 39 and 374. All patients in the ad hoc analysis were treated with evocalcet and, therefore, further studies are needed to answer the reviewer’s comment. 

Introduction. The statement that IV administration of VDRA « is the standard therapy for patients with SHPT » is incorrect as such. At least in the Western world VDRA can by given either orally or intravenously for the treatment of SHPT.

Response: Thank you for pointing this out; however, unlike Western countries, IV VDRA is the standard therapy for patients with SHPT and oral VDRA is used to supplement VD and/or treat hypocalcemia in Japan. To clarify the different application of oral VDRA, “in Japan” has been added in Line 57. 

Discussion, lines 238-284. The statement is inappropriate that « intact PTH levels were slightly lower in the LDG and HDG compared with the NDG » since the difference was not statistically significant.

Response: The sentence “In this study, we found that over the 30 weeks of treatment, intact PTH levels were slightly lower in the LDG and HDG compared with the NDG” has been corrected to “Although it was not statistically significant, we found that over the 30 weeks of treatment, intact PTH levels tended to be slightly lower in the LDG and HDG compared with the NDG” in Line 303 to emphasize that it was not statistically significant. 

Reviewer #2: 

This study has demonstrated that the association of oral evocalcet with vitamin D activator was beneficial, with lower synthesis of FGF-23, lower incidence of hypocalcemia and with more patients achieving the calcium target. This oral formulation of evocalcet, was also associated with a reduction of gastric symptoms when compared with oral cinacalcet.

This study confirms the benefit of the treatment of hyperparathyroidism with an association of a calcimimetic + vitamin D.

The reduction of gastric symptoms verified with evocalcet will contribute for a better adherence of this new oral treatment for hyperparathyroidism.

However, the intravenous administration of etelcalcetide after hemodialysis, assures the patient adherence to this kind of treatment.

Evocalcet will be very useful for the treatment of patients with hyperparathyroidism who develop gastric symptoms associated with cinacalcet and who have no conditions for a three times/week intravenous treatment. This will be the case, for instance, of DP patients.

Response: We appreciate the reviewer’s comments on etelcalcetide and its comparison with evocalcet regarding its adherence and applications. Both drugs have advantages and disadvantages that can be used to provide the best available treatment according to each patient’s clinical characteristics and needs. In the previous head-to-head comparison study, the efficacy and safety of oral cinacalcet and evocalcet were compared and, therefore, we would like to keep the discussions on the comparisons between these two oral calcimimetics in this study. 

Reviewer #3: This paper reported an ad-hoc analysis results using patients enrolled in one treatment arm of a completed phase III trial. Patients were categorized into three groups according to average weekly VDRA dose. Many clinical outcomes and adverse drug reactions were compared among these three groups. However, almost all statistical analysis only employed basic statistical tests and multiple testing adjustment was not implemented. In addition, most statements about the dose effect of VDRA were without rigorous statistical evidences presented. A professional statistician is recommended to be involved for rigorous data analysis and drawing conclusions.

Specifically,

1. Figure 1 displayed longitudinal mean measurements for each group of patients. It would be helpful to also include variations in the plots. In addition, linear mixed models are recommended to do a formal comparison among three patients groups and make conclusions rather than just looking at the plots and stating there was a tendency of VDRA dose-dependent trend.

Response: Standard deviations have been summarized in S2 Table. We appreciate the reviewer’s comment regarding the statistical analysis. The original phase 3 study, as well as this ad hoc analysis, was designed to statistically test the guideline target achievement rate among the treatment groups for mineral and bone disorder parameters, but not to test significant longitudinal trends in the repeated measures within each group or those among the groups. We are aware that a linear mixed model is a useful statistical tool whereby 30 weeks of measurements can be tested among the three groups unlike ANOVA. We will consider the reviewer’s comment when we design future studies. Please see our response to reviewer 1 comment 4 regarding the lack of statistical analyses on the VDRA dose-dependent trends. 

2. Table 1 implies that these three groups of patients were not balanced in all baseline measurements. Therefore, the corresponding result comparisons should consider adjusting for these possible confounding factors.

Response: As the reviewer pointed out, we considered adjusting the outcome data with confounding factors, such as age, and baseline FGF23 and BAP levels. However, because of the small sample size (n=45 in the LDG group) it was impossible to adjust for such factors. Therefore, we have stated the potential for bias in Line 353. 

3. A conclusion from Figure 4 says "The slope of the FGF23 intact and C-terminal relationship increased after treatment with evocalcet at Week 30 compared with Week 0 in all patients, as well as by VDRA DG." First, the estimated slope were not reported. Second, a formal statistical analysis should be implemented here to provide support of such statement.

Response: The estimated slopes for Week 0 and Week 30 in all patients, and those in three DGs, have been added to Fig. 4 (now changed to Fig. 5 after revision). The statistical analysis plan was created and approved for this study and, therefore, it is difficult to run an additional statistical analysis. Furthermore, because of the small sample size, we planned to find and discuss any tendency rather than statistical significances in this study.

Minor comments:

1. The title of Table 2 says "Mean numbers and percentages...". However, outcomes in Table 2 are all binary ones. It is confusing why there would be mean numbers and percentages.

Response: “Mean” has been deleted. 

2. It is also helpful to add in variation of those ratios to Figure 3.

Response: Error bars have been added to Fig. 3 (now changed to Fig. 4 after revision).

---

## [Decision Letter · Decision Letter 1]

24 Jun 2021

PONE-D-21-07286R1

Evocalcet with vitamin D receptor activator treatment for secondary hyperparathyroidism

PLOS ONE

Dear Dr. Shigematsu

Thank you for submitting your manuscript to PLOS ONE. After careful consideration, we feel that it has merit but does not fully meet PLOS ONE’s publication criteria as it currently stands. Therefore, we invite you to submit a revised version of the manuscript that addresses the points raised during the review process.

We look forward to receiving your revised manuscript.

Kind regards,

Pasqual Barretti, Ph.D., MD

Academic Editor

PLOS ONE

Additional Editor Comments (if provided):

The questions on statistical aspects from the reviewer 3 ara relevant and need to be address.

Reviewers' comments:

Reviewer's Responses to Questions

**Comments to the Author**

1. If the authors have adequately addressed your comments raised in a previous round of review and you feel that this manuscript is now acceptable for publication, you may indicate that here to bypass the “Comments to the Author” section, enter your conflict of interest statement in the “Confidential to Editor” section, and submit your "Accept" recommendation.

Reviewer #1: (No Response)

Reviewer #3: (No Response)

2. Is the manuscript technically sound, and do the data support the conclusions?

Reviewer #1: Yes

Reviewer #3: Partly

3. Has the statistical analysis been performed appropriately and rigorously? 

Reviewer #1: Yes

Reviewer #3: No

4. Have the authors made all data underlying the findings in their manuscript fully available?

Reviewer #1: Yes

Reviewer #3: Yes

5. Is the manuscript presented in an intelligible fashion and written in standard English?

Reviewer #1: Yes

Reviewer #3: Yes

6. Review Comments to the Author

Reviewer #1: The authors have provided satisfactory answers to all my queries and modified the manuscript as appropriate.

Reviewer #3: Previous comments related to rigorous statistical analysis were not fully addressed and hence this manuscript still does not reach the standard for publication. Authors are expected to fully address previous statistical comments again.

This is a post ad hoc analysis, so it is not convincible that no additional statistical analysis could be added because of a previously approved statistical analysis plan which is not submitted with this manuscript at all. There is one group of patients having relatively low number of patients (N=45), however, this sample size is not too small to adjust for any covariate. In addition, for a rigorous inference following any statistical analysis, 95% CI should be reported along with point estimate all the time. For example, the estimated slopes reported in Figure 5 should be reported along with the corresponding 95% CI.

7. PLOS authors have the option to publish the peer review history of their article (what does this mean?). If published, this will include your full peer review and any attached files.

Reviewer #1: No

Reviewer #3: No

---

## [Author Response · Author response to Decision Letter 1]

13 Oct 2021

Below are our point-by-point responses to all the reviewers’ comments.

Reviewer #3: Previous comments related to rigorous statistical analysis were not fully addressed and hence this manuscript still does not reach the standard for publication. Authors are expected to fully address previous statistical comments again. 

Response: We have addressed all previous statistical comments. Please see our responses below.

This is a post ad hoc analysis, so it is not convincible that no additional statistical analysis could be added because of a previously approved statistical analysis plan which is not submitted with this manuscript at all. There is one group of patients having relatively low number of patients (N=45), however, this sample size is not too small to adjust for any covariate. In addition, for a rigorous inference following any statistical analysis, 95% CI should be reported along with point estimate all the time. For example, the estimated slopes reported in Figure 5 should be reported along with the corresponding 95% CI.

Response: Please see our response to the third comment below. 

Previous statistical comments

1. Figure 1 displayed longitudinal mean measurements for each group of patients. It would be helpful to also include variations in the plots. In addition, linear mixed models are recommended to do a formal comparison among three patients groups and make conclusions rather than just looking at the plots and stating there was a tendency of VDRA dose-dependent trend.

Response: Differences in changes in intact PTH, and corrected calcium and phosphorus from baseline to Week 30 among the three treatment groups were analyzed with a repeated linear mixed model, as suggested. As you will find in the Results section on pages 12–13, significant differences in the intact PTH, cCa, and phosphorus levels were found between certain groups. However, no significant difference was observed for the intact PTH, cCa, or phosphorus level among the three dose groups when the levels were adjusted according to their respective baseline levels (all cases p > 0.95). These results therefore suggest that evocalcet exerts its effect regardless of VDRA use or its dose.

2. Table 1 implies that these three groups of patients were not balanced in all baseline measurements. Therefore, the corresponding result comparisons should consider adjusting for these possible confounding factors.

Response: We have added the between-group differences of the target parameter achievement rates adjusted for age (< 65 and ≥ 65 years), sex, and intact PTH level (< 500 pg/mL and ≥ 500 pg/mL) at baseline and previous cinacalcet treatment to Table 2. 

3. A conclusion from Figure 4 says "The slope of the FGF23 intact and C-terminal relationship increased after treatment with evocalcet at Week 30 compared with Week 0 in all patients, as well as by VDRA DG." First, the estimated slope were not reported. Second, a formal statistical analysis should be implemented here to provide support of such statement.

Response: As suggested by the reviewer, changes in the ratio of intact-to-C-terminal FGF23 between Week 0 and Week 30 were analyzed with a generalized linear mixed model. As shown below, the slope of the FGF23 intact and C-terminal relationship increased after treatment with evocalcet at Week 30 compared with Week 0 in all patients (p = 0.0002) and patients in the NDG (p = 0.0005). These results suggest a reduction in FGF23 synthesis. Although no statistical significance was observed, the slopes tended to increase in the LDG (p = 0.0696) and HDG (p = 0.0741). Results of the statistical analysis have been included in the Results section on page 19.

---

## [Decision Letter · Decision Letter 2]

6 Jan 2022

Evocalcet with vitamin D receptor activator treatment for secondary hyperparathyroidism

PONE-D-21-07286R2

Dear Dr. SHIGEMATSU

We’re pleased to inform you that your manuscript has been judged scientifically suitable for publication and will be formally accepted for publication once it meets all outstanding technical requirements.

Kind regards,

Pasqual Barretti, Ph.D., MD

Academic Editor

PLOS ONE

Additional Editor Comments (optional):

Actually, all of the invited reviewer in different steps of this revision process, have decided to accept the manuscript. In my opinion the manuscript has improved since its original submission. Therefore, may decision is " accept"

Reviewers' comments:

Reviewer's Responses to Questions

**Comments to the Author**

1. If the authors have adequately addressed your comments raised in a previous round of review and you feel that this manuscript is now acceptable for publication, you may indicate that here to bypass the “Comments to the Author” section, enter your conflict of interest statement in the “Confidential to Editor” section, and submit your "Accept" recommendation.

Reviewer #3: All comments have been addressed

2. Is the manuscript technically sound, and do the data support the conclusions?

Reviewer #3: Yes

3. Has the statistical analysis been performed appropriately and rigorously? 

Reviewer #3: Yes

4. Have the authors made all data underlying the findings in their manuscript fully available?

Reviewer #3: Yes

5. Is the manuscript presented in an intelligible fashion and written in standard English?

Reviewer #3: Yes

6. Review Comments to the Author

Reviewer #3: This time I agree that all my previous comments are fully addressed. So I recommend to accept it now.

7. PLOS authors have the option to publish the peer review history of their article (what does this mean?). If published, this will include your full peer review and any attached files.

Reviewer #3: No